# A Novel Electrochemical Sensor Based on an Environmentally Friendly Synthesis of Magnetic Chitosan Nanocomposite Carbon Paste Electrode for the Determination of Diclofenac to Control Inflammation

**DOI:** 10.3390/nano13061079

**Published:** 2023-03-16

**Authors:** Mohamed Abd-Elsabour, Mortaga M. Abou-Krisha, Sayed H. Kenawy, Tarek A. Yousef

**Affiliations:** 1Chemistry Department, Faculty of Science, South Valley University, Qena 83523, Egypt; 2Chemistry Department, College of Science, Imam Mohammad Ibn Saud Islamic University (IMSIU), Riyadh 11623, Saudi Arabia; 3Refractories, Ceramics and Building Materials Department, National Research Centre, El-Buhouth St., Dokki, Giza 12622, Egypt; 4Mansoura Laboratory, Toxic and Narcotic Drug, Forensic Medicine Department, Medicolegal Organization, Ministry of Justice, Cairo 11435, Egypt

**Keywords:** magnetic chitosan nanocomposite, diclofenac, cyclic voltammetry, differential pulse voltammetry, chronoamperometry

## Abstract

A simple and eco-friendly electrochemical sensor for the anti-inflammatory diclofenac (DIC) was developed in a chitosan nanocomposite carbon paste electrode (M-Chs NC/CPE). The M-Chs NC/CPE was characterized with FTIR, XRD, SEM, and TEM for the size, surface area, and morphology. The produced electrode showed a high electrocatalytic activity to use the DIC in 0.1 M of the BR buffer (pH 3.0). The effect of scanning speed and pH on the DIC oxidation peak suggests that the DIC electrode process has a typical diffusion characteristic with two electrons and two protons. Furthermore, the peak current linearly proportional to the DIC concentration ranged from 0.025 M to 4.0 M with the correlation coefficient (r2). The sensitivity, limit of detection (LOD; 3σ), and the limit of quantification (LOQ; 10σ) were 0.993, 9.6 µA/µM cm^2^, 0.007 µM, and 0.024 µM, respectively. In the end, the proposed sensor enables the reliable and sensitive detection of DIC in biological and pharmaceutical samples.

## 1. Introduction

Diclofenac sodium (DIC) is an anti-inflammatory non-steroidal agent with analgesic and anti-pyretic properties [1]. DIC is used to treat many severe inflammatory and painful diseases such as rheumatoid arthritis, osteoarthritis, soft tissue disorders, renal ulcers, acute gout, diabetes, and migraine [2,3]. Research shows that DIC is the most effective and most assertive NSAID. Additionally, various studies on the use of DIC have proved it efficacy with fewer side effects in terms of quality assurance compared to most other NSAIDs [1,2,3]. On the other hand, DIC can cause life-threatening problems such as heart attacks and stroke. Furthermore, it can harm people and cause aplastic anemia, gastrointestinal problems, agranulocytosis, and abnormal renal function [3]. Thus, the determination of the DIC trace levels in the biological and pharmaceutical samples is crucial for development and treatment. To date, a wide range of methods have been employed to detect DIC in different kinds of materials such as spectrophotometry [4,5], fluorimetry [6], HPLC [7,8], GCMS [9,10], LCMS [11] electrokinetic chromatography [12], TLC [13], and electrochemical methods [14,15,16,17,18,19,20,21,22,23]. Among these numerous analytical techniques, the electrochemical approaches might be thought of as the best ones for DIC measurement in solutions since they are sensitive, selective, cheap, and simple to use.

Carbon paste electrodes (CPEs) are used in various electrochemistry applications for their ease of preparation, speed, response stability, porous surface, renewal, low cost, low ohmic resistance, and lack of internal solutions [24]. Recently, the use of nanocomposites based on biopolymers to improve the efficiency, sensitivity, and electrocatalytic activity of various electrochemical sensors due to the presence of several active sites such as the hydroxyl, carboxyl, and amino groups included in biopolymer, which may interact with different analytes via the formation of electrostatic forces or hydrogen bonds [25]. Moreover, the nanoscale size of the composites revealed a clear impact on the increase in the surface area of the electrode [26].

Chitosan (Chs) is considered the second most prevalent naturally occurring biopolymer in the world. It can be produced from the deacetylation process of chitin obtained from natural wastes such as shellfish and crustaceans [27]. Chs has a variety of intriguing applications including organo catalysis [28], supramolecular polymer networks [29], drug delivery systems [30], treatment of water [31], and biosensors [32] due to its various advantages such as the ability to be extracted from waste sources with low cost, nontoxic, and possesses attractive properties involving biocompatibility, biodegradability, hydrophilic, and basic moieties [33].

Subsequently, a simple, eco-friendly, and cheap protocol was used for the synthesis of magnetic chitosan nanocomposite (M-Chs NC) through the extraction of nano-chitosan from natural waste sources (shrimp shell) followed by its hybridization with magnetite depending on the coprecipitation method. The obtained nanocomposite possesses versatile properties such as compatibility, low-cost, and superparamagnetic behavior, which provides a strong motive for its exploitation in the design of a stable interface, rapid, and sensitive modified CPE for the detection of DIC for the first time. The fabricated electrode showed high performance toward DIC in real biological and pharmaceutical samples.

## 2. Experimental Procedures

### 2.1. Materials

All chemicals used were of an analytical quality and used without further purification; HCl (36.0%), NaOH (≥98.0%, pellets (anhydrous)), acetone (≥99.8%), NaOCl (available chlorine 10–15%), Tri polyphosphate (TPP) (85.0%), acetic acid (95.0%), FeSO_4_·7H_2_O (≥99.0%), FeCl_3_·6H_2_O (≥99.0%), NH_4_OH (25.0%), and DIC (≥98.0% purity) were supplied from Sigma–Aldrich Co., Mumbai, Maharashtra, India. A DIC stock solution was prepared by dissolving the calculated amount in 100 mL double distilled water and then diluting the DIC concentration required. The prepared stock solutions were stored in the refrigerator until they were used in the laboratory. Several freshly prepared electrolytes (El Nasr Pharm. Chem. Co., Cairo, Egypt) such as KCl, acetate buffer solution (ABS), phosphate buffer solution (PBS), BR buffer, and borate buffer solution (BBS) were used. Double distilled water was employed to prepare aqueous solutions at room temperature.

### 2.2. Instruments

FTIR has been employed as a quick and accurate method to provide information about the chemical composition of the prepared samples and confirm the interaction between Chs and magnetite NPs. Thus, the KBr pellet method was employed to examine the FTIR spectra of the prepared samples in the range of 400–4000 cm^−1^ by using a Nicolet 5700 infrared spectrometer (Nicolet, MA, USA). A powder X-ray diffractometer (Brucker D8 Advance, Karlsruhe, Germany with a Cu Kα radiation source, λ = 1.5406 Å, in the range of 2θ = 10–80° using a step size of 0.02/min) was used to record the X-ray diffraction spectra of the prepared samples for the investigation of the chemical composition and phase structure of the samples. The Scherrer equation (D = 0.9λ/(βcos θ)), where D is the average crystallite size (nm), λ is the X-ray wavelength used, θ is the diffraction angle, and β is the full width at half the maximum of the diffraction peak in radians, was used to calculate the average crystalline size of the prepared sample. To examine the topographical, morphological, and compositional features of the produced samples, SEM (Jeol, JSM-IT200, Shinagawa Seaside, Tokyo, Japan) and TEM (Jeol Jem-1230 Shinagawa Seaside, Tokyo, Japan) operating at an accelerating voltage of 30–200 kV was used. A Potentiostat 263 (EG G PARC) from the Princeton Applied Research Corporation, manufactured in the United States (Oak Ridge, TN, USA), was connected to the PC by 352 corrosion software and used to quantity the electrochemical measurements. The electrochemical cell used (MODEL K0264 MICRO-CELL) consisted of three electrodes: an auxiliary platinum wire electrode (Model K0266) and an Ag/AgCl electrode saturated with a 3 M KCl reference electrode (Model K0265). In addition, M-Chs NC modified CPE was used as a working electrode.

### 2.3. Preparation of Chs NPs

Shrimp shell waste was collected from a local fish market and used as a cheap source to extract chitosan. The collected shrimp shells were thoroughly washed with distilled water to remove all soluble organic and inorganic materials, then the cleaned shells were heated for about 20 min at 90–100 °C to inactivate the endogenous enzymes. The shells were dried and ground into a fine powder followed by the demineralization of the shrimp shell powder, which primarily involves the removal of calcium carbonate along with other elements, and was carried out through stirring the shell powder with a HCl solution (1N) with ratio of 1:30 (w/v) for 75 min at room temperature, then filtering, washing with distilled water to neutrality and drying. Deproteinization was used to remove adhering protein from the demineralized shrimp shell powder by combining it with NaOH solution (3N) for 75 min at room temperature, then filtering, washing with distilled water to neutrality, and drying to provide chitin. The obtained product was treated with acetone and sodium hypochlorite to remove pigments such as carotenoids and melanin. This was followed by the deacetylation step of bleaching chitin in which chitosan was obtained by the alkaline hydrolysis treatment of 1 g of chitin with 50 mL of sodium hydroxide (50%) for about 50 min at 90 °C. The product was filtered, thoroughly washed with distilled water and then treated with ethanol followed by drying for 24 h at 80 °C. Finally, the extracted Chs was treated with acetic acid solution (2%) for 30 min, followed by the addition of 40 mL of tripolyphosphate to 100 mL of extracted Chs solution, and stirring for 2 h to obtain Chs nanoparticles that were separated by centrifugation, then rinsed with distilled H_2_O and dried (see Figure 1) [34,35,36].

### 2.4. Synthesis of M-Chs NC

Using the co-precipitation approach, M-Chs NC was prepared (Figure 2). First, 1 g of the extracted Chs was stirred magnetically with 100 mL of Fe^2+^/Fe^3+^ solution (molar ratio 1:2, respectively) for 3 h at 50 °C. Subsequently, the resulting mixture was ultrasonicated for 30 min at 100 °C. The formation of magnetite was accomplished by the dropwise addition of an equivalent amount of 25% ammonia solution. After the completion of the reaction, the obtained black precipitate was separated by a permanent magnet and washed with doubly distilled water several times until neutralization. Finally, the obtained sample was dried in an electric oven at 80 °C overnight.

### 2.5. Preparation of Pharmaceutical Samples

Voltaren^TM^ (100 mg) was used as the DIC pharmaceutical samples. Five tablets were taken and ground into a fine powder using a mortar and pestle. The required amount of DIC was dissolved in a suitable volume of BR (pH 3.0) to obtain a specific concentration of DIC. Then, the resulting mixture was sonicated for 10 min followed by filtration to obtain the pure solution and then stored in the refrigerator for the next use.

### 2.6. Preparation of Human Serum Samples

Blood from healthy volunteers was collected in order to produce the serum sample. The serum was extracted from the samples by centrifuging them at 5000 rpm for 10 min. Then, to precipitate any protein that might be present in the sample, 2 mL of the resulting serum was added to another tube that contained 4 mL of acetonitrile. The tube was vortexed for 10 min before being centrifuged at 5000 rpm for 20 min. The electrochemical cell received the supernatant without delay. The last step involved adding aliquots of DIC and diluting them with BR buffer (pH = 3.0) until the desired DIC concentrations were prepared.

## 3. Results and Discussion

### 3.1. Characterization of Extracted Chs NPs and M-Chs NC

#### 3.1.1. Structural Analysis

The present research revealed that M-Chs NC was synthesized using an in situ co-precipitation method. The FTIR spectra of Chs NPs and M-Chs NC were performed to explore the interaction between Chs NPs and Fe_3_O_4_ NPs in the prepared NC (Figure 3a). The FTIR spectrum of the pure extracted Chs exhibited several transmitted bands. The stretching vibrations of the OH and NH_2_ groups were related to the strong broad band at 3421 cm^−1^. The absorption bands at 2926 and 2876 cm^−1^ were characteristic of the asymmetric and symmetric C–H aliphatic stretching vibrations. C=O stretching vibration and N–H bending vibration were attributed to the bands at 1640 and 1552 cm^−1^, respectively. The bands at 1385 and 1073 cm^−1^ were assigned to the bending vibration of C–N and the stretching vibration C–O groups, respectively [37]. On the other hand, the FTIR spectrum of M-Chs NC showed a match in the same absorption bands as those of the pure Chs with little change in both the band position and intensity. The bands at 3421, 2926, 1640, 1385, and 1073 cm^−1^ shifted to 3412, 3140, 1630, 1398, and 1114 cm^−1^, respectively. These results demonstrate the complexation of Fe^3+^ with the included amino and hydroxyl groups on Chs NPs. Moreover, the appearance of two strong characteristic adsorption bands at 606 and 426 cm^−1^ were due to the stretching vibrations of Fe–O of Fe_3_O_4_, which indicates the in situ co-precipitation of Fe_3_O_4_ on Chs NPs [38].

The crystal structure and crystallite size of the produced samples were characterized using the XRD technique. At room temperature, the XRD spectra of the prepared samples were recorded in the range of 10° to 70°. The semi-crystalline nature of the extracted Chs NPs was revealed by the presence of a characteristic peak at 2θ = 20.01° in its XRD pattern (Figure 3b). The XRD study revealed the crystallinity of chitosan from shrimp shells as well as the appearance of a significant chitosan peak at high intensity [39]. Furthermore, the absence of any other diffraction peaks related to contaminants demonstrated their purity. Otherwise, the peak corresponding to the extracted Chs was shifted to approximately 21.2° and became wider in the XRD pattern of as-prepared M-Chs (Figure 3b). Furthermore, the M-Ch NC XRD pattern exhibited six characteristic diffraction peaks of cubic Fe_3_O_4_ NPs at 2θ = 30.2°, 35.5°, 43.2°, 53.7°, 57.2°, and 62.8° corresponding to Miller indices of (220), (311), (400), (422), (511), and (440), respectively. Each of the diffraction peaks were observed in a similar position to those of magnetite (JCPDS File No. 19-0629) [40]. The high purity of the prepared magnetite was confirmed by the absence of any other peaks of any other phases in the XRD pattern. Moreover, the strong diffraction peaks indicate the high crystalline nature of the prepared sample. Using the Debye–Scherrer formula and the analysis of the XRD data of (311) and (511), it was determined that the average crystalline size of Fe_3_O_4_ NPs was around 17 nm.

#### 3.1.2. Morphological Analysis

The microstructure as well as the particle size of the prepared samples were investigated using SEM at a magnification of X60000 and TEM. Figure 4a,b shows that the SEM micrographs of the shrimp shell Chs NPs had spherical and smooth surface nanoparticles with a uniform diameter of approximately 80 nm. The higher surface area of Chs NPs was confirmed by its spherical shape, which allowed for a successful loading of Fe_3_O_4_ on its surface [41]. The SEM of M-Chs NC (Figure 4b) showed small, agglomerated particles, which is like that of Chs NPs, with slight differences in the particle size. These particles had a quasi-spherical shape with a rough and irregular surface, which is characteristic for typical hybrid materials. The extracted Chs NPs were well-dispersed quasi-spherical and had a diameter around 50 nm, as seen in Figure 4c. Based on the obtained results, the authors designed a feasible, low-cost method for producing Chs NPs that were less than 100 nm in size, in contrast to other methods. The TEM image of M-Chs NC shows the core-shell structure of Fe_3_O_4_ NPs (Figure 4d). The pictures showed the core of the Fe_3_O_4_ NPs, which was dark in color. These differed from the Chs, which appeared as the grey coating surrounding it. The average size of the nanocomposites was greater than 80 nm.

### 3.2. Electroactive Surface Area Measurements

In the bare CPE, Chs NPs/CPE and M-Chs NC/CPE with a scan rate of 50 mV/s, the cyclic voltammograms (CVs) of 1.0 mM of [Fe(CN)_6_]^3−/4−^ in the presence of KCl (0.1 M) as a supporting electrolyte are shown in Figure 5. According to the [Fe(CN)_6_]^3−/4−^ findings, a reversible redox reaction with separation peak potentials (ΔEp) of 560 mV, 280 mV, and 130 mV for the bare CPE, Chs NPs/CPE, and M-Chs NC/CPE, respectively, took place at the surface of all applied electrodes. Additionally, the anodic peck current at M-Chs NPs/CPE was enhanced by 100 A and 400 A, respectively, compared to the bare CPE at Chs NPs/CPE and M-Chs NC/CPE. This result suggests that the M-Chs NPs enhanced the electrochemical signal at the modified electrodes and decreased the charge-transfer resistance to boost the CPE’s electrochemical response. The M-Chs NC/CPE’s electrocatalytic activity findings show that it may be used for suitable analytical applications. The active surface area (A) for the fabricated electrodes was calculated using the Randles–Sevcik formula (I_p_ = (26.9 × 10^4^) n^1.5^ A D_R_
^0.5^ υ^0.5^ C_o_), wherever D_R_ is the diffusion coefficient (for [Fe(CN)_6_]^3−/4−^ D_R_ = 7.6 × 10^−6^ cm^2^ s^−1^)), I_p_ denotes the maximum current, and n is the number of transferred electrons involved in the electrochemical process (n = 1) [41]. The value of A was found to be 0.008 cm^2^, 0.012 cm^2^, and 0.035 cm^2^ for unmodified CPE, Chs NPs/CPE, and M-Chs NC/CPE, respectively, indicating the significant increase in the electroactive surface area of M-Chs NC/CPE with CPE.

### 3.3. Electrochemical Behavior of DIC

CV was applied to study the electrochemical performance of 0.5 µΜ DIC at the surface of different applied working electrodes in 0.1 M of BR buffer (pH 3.0) in the potential range of 0.2–1.1 V at a scan rate of 50 mV s^−1^. As shown in Figure 6a, a low redox signal of DIC was attained in the bare CPE (curve a). Meanwhile, in the case of Chs NPs/CPE (curve b) and M-Chs NC/CPE (curve c), the anodic peak current signal increased by 2.04 and 4.0 times compared with that obtained at the surface of the unmodified CPE. It was also found that the potential value decreased by around 0.088 V, which shows that the applied modifier promoted the electrochemical process and markedly accelerated the rate of electron transfer. The electrocatalytic efficiency of the modified electrode can be clarified by the high excellent adsorption capability and catalytic action of M-Chs NC. Therefore, M-Chs NC was anticipated to increase the accumulation of DIC particles on the altered electrode surface and increase the active surface area of the CPE [37]. To examine the impact of the adsorption of DIC oxidation product on the voltammetric response of DIC at the M-Chs NC/CPE surface, two continuous scan cycles were recorded. According to Figure 6b, the first cycle had a clearly defined oxidation peak at about 0.81 V, which in the reverse scan developed into a cathodic peak at about 0.56 V. The second cycle, however, revealed the appearance of a new oxidation peak at 0.58 V that coupled reversibly with the cathodic peak at 0.3 V. The redox characteristic of the oxidation product of DIC caused the reversible pair to develop at less positive potentials. Additionally, a decrease in the anodic peak current of DIC was observed in the second cycle because of the high adsorption capacity of the oxidation products of DIC at the M-Chs NC/CPE surface, which blocks the electroactive sites at the electrode surface. According to previously published data, DIC is irreversibly oxidized and the donation increases to an oxidation peak at 0.81 V when the potential sweep is started in the positive direction at the surface of M-Chs NC/CPE [1,2].

### 3.4. Effect of Scan Rate

The scan rate was used to study the influence of the potential on the nature of the electro-oxidation process of DIC at M-Chs NC/CPE, and linear sweep voltammetry (LSV) for 0.5 μM of DIC was applied at various scan rates (Ʋ) in 0.1 M of BR buffer (pH = 3.0). According to the computed results, increasing the scan rate from 0.01 to 0.75 V/s caused the anodic peak potentials of the DIC to shift toward positive values, as shown in Figure 7a. The current signal also grew as the scan speeds increased. The derived linear relationship between I_p_ and Ʋ^0.5^ (Equation (1)) (inset Figure 7a) shows that the DIC oxidation process is diffusion-based [42]. Additionally, charting the relationship between log I_p_ and the log Ʋ, as shown in Figure 7b, revealed that the diffusion nature of DIC’s oxidation process resulted in a straight line that followed Equation (2). The calculated slope of 0.6 was extremely close to the calculated electron transfer coefficient (α), which was assumed to be equal to 0.5 [43] for the diffusion-controlled process.

The total number of electrons involved in the DIC oxidation process (n) was calculated using Laviron equation, Equation (3). The value (αn) was calculated from the slope of the relationship between *E_p_* and log Ʋ, as shown in Figure 7c and Equation (4). The DIC oxidation process therefore involves two electrons, and the predicted mechanism can be expressed as shown in (Figure 1) [2,3].
(1)Ip µA=−2.98+48.96 Ʋ0.5 (Vs)0.5 r2=0.997
(2)log Ip µA=1.67+0.60 log Ʋ Vs r2=0.996
(3)ΔEpΔlogƲ=0.059 αn
(4)Ep V=0.88+0.051 log Ʋ Vs r2=0.996

### 3.5. Effect of pH

To study the influence of the pH on the anodic response of DIC at M-Chs NC/CPE, the LSV of 0.5 µM DIC was recorded at different pH values of the BR buffer ranging from 3.0 to 7.0 (Figure 8a). It was found that as the pH increased, the peak potential shifted toward higher negative values because of the proton’s involvement in the oxidation reaction, as seen in Figure 5b. The Ep versus pH plots showed a linear connection (Figure 8b), which is supported by Equation (5).
(5)$$Ep\;\left(V\right)=098-0.047\;\mathrm{pH}\;r^{2}=0.997$$

For electrochemical processes involving the same number of protons and electrons, the slope of the equation was quite close to the predicted Nernstian value of 0.059 V/pH [42]. This conclusion is in accordance with Figure 1 (probable oxidation mechanism of DIC), which was previously described. Additionally, when the pH value increased from 3.0 to 7.0, the anodic peak current of the DIC gradually decreased, and when it further increased up to pH 7.0, it caused the decline in the current signal. In turn, the electrochemical oxidation of hydrolytic products may be used to explain the emergence of a new peak that was only visible at a higher pH [44]. At pH 3.0, the highest peak was observed, thus, pH 3.0 was considered the optimal value for all experiments.

### 3.6. Chronoamperometric Study

The chronoamperometric method was utilized to estimate the diffusion coefficient (D) value for the voltammetric oxidation of DIC on the M-CHS NC/CPE surface using the Cottrell equation [45] (Equation (6)).
(6)$$Ip=n\;F\;A\;D^{1/2}C\;\pi^{-1/2}t^{-1/2}$$
where *A* represents the geometric surface area of M-CHS NC/CPE; C is the DIC concentration (mM); and *t* symbolizes the time elapsed (s). Figure 9 illustrates the chronoamperograms of different concentrations of DIC (0.05, 0.075, and 0.1 µM) at a constant potential of 0.82 V in 0.1 M of BR buffer (pH = 3.0). For different DIC concentrations, the relationship between I and *t*^−1/2^ produced straight lines, and the diffusion coefficient was discovered to be 2.70 × 10^−5^ cm^2^/s.

### 3.7. Calibration Curve and Detection Limit

The impact of the DIC concentration on the oxidation peak current across the range of 0.025 µM to 4.0 µM was examined using differential pulse voltammetry (DPV). It was discovered that I_p_ increased proportionately to the DIC concentration under optimal conditions (as depicted in Figure 10), resulting in a linear calibration relationship and the linear angle coefficient was equal to 0.9998.
(7)$$Ip\;\left(µA\right)=4.55+9.60\;C\;\left(µM\right)\;r^{2}=0.993$$

The limit of detection (LOD) and quantification (LOQ) were determined to be 0.007 µM and 0.024 µM, respectively, utilizing the formulas LOD = 3 s/m and LOQ = 10 s/m, where s represents the standard deviation of the peak current of the lowest concentration of the linearity range and m represents the slope of the corresponding calibration equation [43]. The outcomes demonstrated the significant sensitivity of the modified sensor toward DIC oxidation. The LOD for DIC detection using variously tuned electrodes is compared in Table 1.

### 3.8. Effect of Interferences

To evaluate the capability of M-Chs NC/CPE for the sensitive detection of DIC in the presence of various impurities or matrix components, the effect of an increase in the concentrations of the probable interferences on the oxidation peak of 1.0 μM DIC was examined. In the presence of about 500-fold excess, K^+^, Na^+^, Fe^3+^, SO_4_^2−^, Cl^−^, and some additives present in the tablet compositions such as magnesium stearate, starch, cellulose, and talc. Additionally, at about a 50-fold excess of uric acid, dopamine, glucose, sucrose, citric acid, and ascorbic acid, there was no significant influence on the DIC oxidation peak. Therefore, the proposed electrode can be highly effective in selectively detecting DIC in real samples.

### 3.9. Repeatability and Stability

In order to evaluate the repeatability of M-Chs NC/CPE, the net response of the modified electrode was monitored in six duplicates in 1.0 M DIC. The relative standard deviation (RSD) for the six subsequent assays was 2.04%. Additionally, the stability of the sensor storage was evaluated; according to the findings, after 21 days of room temperature storage, M-Chs NC/CPE lost approximately 7.1% of its initial response (25 °C). Thus, the repeatability and stability of the M-Chs NC/CPE are acceptable.

### 3.10. Applications

The standard addition method was applied to study the applicability of utilizing M-Chs NC/CPE for the detection of DIC in real samples. The results obtained under the optimal conditions are shown in Table 2. The DIC voltammetric measurements of DIC at M-Chs NC/CPE indicate satisfactory precision with recoveries of 97.0 to 102.7%. As a result, M-Chs NC/CPE can be successfully utilized for the sensitive sensing of DIC in real samples.

## 4. Conclusions

A simple and costless synthesis of M-Chs NC was employed to further improve the properties of the CPE. The physical properties of the prepared M-Chs NC/CPE were characterized by FTIR, XRD, SEM, and TEM. M-Ch NC/CPE showed favorable electro-catalytic behavior toward the [Fe(CN)_6_]^3−/4−^ redox couple with the highest electroactive surface area (0.035 cm^2^). High electrocatalytic oxidation of DIC in 0.1 M of BR buffer (pH 3.0) was achieved on the M-Chs NC/CPE surface. The peak current of the DIC increased with an increasing scan rate (0.01–0.750 mV/s) and pH values (3.0–7.0). In addition, a linear dependence of the anodic peak current versus DIC concentration was obtained from 0.025 to 4.0 µM. The value of the diffusion coefficient correlation coefficient, sensitivity, detection limit, and limit of quantification were calculated to be 2.70 × 10^−5^ cm^2^/s, 0.9988, 0.534 µA/µM cm^2^, 0.092 nM, and 0.31 nM, respectively. Finally, the M-Chs NC/CPE is one of the promising electrochemical sensors for the detection of DIC in real samples with high selectivity and stability.

## Data Availability

Data that support the findings of this study are available in this article.

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
