# Peer review of "A Novel Electrochemical Sensor Based on an Environmentally Friendly Synthesis of Magnetic Chitosan Nanocomposite Carbon Paste Electrode for the Determination of Diclofenac to Control Inflammation"

_nanomaterials, 2023, doi:10.3390/nano13061079_

Round 1
Reviewer 1 Report
Manuscript entitled "A Novel Electrochemical Sensor Based on an Environmentally Friendly Synthesis of Magnetic Chitosan Nanocomposite Carbon Paste Electrode for the Determination of Diclofenac to Control Inflammation. is real sound manuscript with good hypothesis. But it requires major corrections as follows or it should be rejected in present form and rewritten after considering the following points.
1) Authors should explain the rational behind using the diclofenac against other NSAIDs
2) What is the experimental evidence to prove the NP stability in serum environment.
3) Fig 4 (a) and (b) size is 2 micro meter, how it is considered as nanometer size.
4) Authrs mentioned NP are stable but SEM images showed the conjugated form with stable, Why ? Scientific clarifications are required.
5) Fig.10. whats the liniear angle coeficient of DIC at BR buffer. more clarifications on experiment and results are required.
6) Line # 393, what is the standard addition method ?
7) IN human serum incubation set , authros found 98% recovery once 15 micro molar added. I wonder how inert are serum protien towards NP. LC-MS data is required.
Author Response
About the Reviewer 1 Comments:
Reviewer #1:
Comments and Suggestions for Authors
Manuscript entitled "A Novel Electrochemical Sensor Based on an Environmentally Friendly Synthesis of Magnetic Chitosan Nanocomposite Carbon Paste Electrode for the Determination of Diclofenac to Control Inflammation. is real sound manuscript with good hypothesis. But it requires major corrections as follows or it should be rejected in present form and rewritten after considering the following points.
1) Authors should explain the rational behind using the diclofenac against other NSAIDs
The Reviewer's remark is completely right, accordingly, more detailes about the importance of DIC were explained in introduction part in the revised manuscript.
2) What is the experimental evidence to prove the NP stability in serum environment.
Deep thanks to the Reviewer for his question, by the application of M-Chs NC/CPE for the detection of DIC in serume environment, the change in the current signal was very small and the obtained results indicated satisfactory precision with recoveries of 97.0 –102.7% as described in part 3.10. That evidence for the stability and high accuracy of the developed electrode in serum media.
3) Fig 4 (a) and (b) size is 2 micro meter, how it is considered as nanometer size.
Deep thanks for the reviwer for his comment, Fig. 4 shows SEM images of chitosan and magnetic chitosan. The SEM bar is 2 µm but it not represent the particle size. When you take a SEM picture from our sample, it means that we can generalize our observations to the overall sample. So that, we can simple count the number of particle shown in the picture and measure their sizes in order to calculate the average size. We use Image J analysis to determine the particle size distribution from SEM images and we found that the average particle size of chitosan was approximately 80 nm as we mentioned in morphological analysis (page 6, line 197). Meanwhile the average particle size of magnetic chitosan was 52.4 nm. The particle size distribution curves of chitosan (a) and magnetic chitosan (b) are attached below. In addition, the nanoscale structures of chitosan and magnetic chitosan in nanometer size are shown in the TEM images (Fig 4c, d) in the revised manuscript.
4) Authrs mentioned NP are stable but SEM images showed the conjugated form with stable, Why ? Scientific clarifications are required.
Thanks for the reviewer for his comment, in the absence of any surface coating, nano-magnetic particles have hydrophobic surfaces with a large surface area to volume ratio. Due to hydrophobic interactions between the particles, they tend to agglomerate forming large clusters. In the present study, SEM image showed the physical aggregation of the nanomagnetic chitosan. There are two possible reasons for this behavior. First, as the Fe3O4 nanoparticles were present in the pores of the chitosan nanoparticles, this may be cross-link between chitosan chains which caused the aggregation of magnetic nanoparticles. Second, the diameter of the particles was too small, and the surface energy was very high, which may cause the physical aggregation (Caspian J Intern Med 2014; 5(3): 156-161). In addition, some Fe3O4 particles recombined, this was due to the magnetic dipole interaction between them. https://doi.org/10.1016/j.apsusc.2013.09.169
5) Fig.10. whats the liniear angle coeficient of DIC at BR buffer. more clarifications on experiment and results are required.
A heartfelt thanks to the reviewer for his question,it was estimated in equation 7 in the revised manuscript.
6) Line # 393, what is the standard addition method ?
Deep thanks to the Reviewer for his question,The method of standard addition is a type of quantitative analysis approach often used in analytical chemistry whereby the standard is added directly to the aliquots of analyzed sample. This method is used in situations where sample matrix also contributes to the analytical signal, a situation known as the matrix effect, thus making it impossible to compare the analytical signal between sample and standard using the traditional calibration curve approach.*
*Harris, Daniel C. (2003). Quantitative Chemical Analysis 6th Edition. New York: W.H. Freeman.
7) IN human serum incubation set , authros found 98% recovery once 15 micro molar added. I wonder how inert are serum protien towards NP. LC-MS data is required.
The Reviewer's remark is completely right, protein may interact with the drug sample thus, during the preparation of human serum samples acetonitrile was added to precipitate any protein as described in part 2.6.
I will very much appreciate receiving any other comments again. Any other suggestions are mostly welcome and please accept all my best wishes.
Sincerely Yours
Sayed H. Kenawy
College of Science,
Chemistry Department,
Imam Mohammad Ibn Saud Islamic University (IMSIU),
Riyadh 11623, KSA
E-mail: skibrahim@imamu.edu.sa

Reviewer 2 Report
M. Abd-Elsabour et al. developed a simple and eco-friendly electrochemical sensor for the anti-inflammatory diclofenac (DIC) by using the chitosan nanocomposite carbon paste electrode (M-Chs NC/CPE). The reviewer feels that not only the researchers involved in sensors but also many readers interested in material chemistry may get inspiration from this report. The reviewer recommends the publication of this manuscript after minor revisions.
1. In Figure 3b, the peak located at around 21o should be clarified. It seems the peak also shows a positive shift. Please provide some explanination.
2. In Figure 4, the font size of the scale bar is not unified.
3. In Figure 6a, the peak intensity and positions of cathodic peaks are also recommended to be described.
Author Response
About the Reviewer 2 Comments:
Comments and Suggestions for Authors
- Abd-Elsabour et al. developed a simple and eco-friendly electrochemical sensor for the anti-inflammatory diclofenac (DIC) by using the chitosan nanocomposite carbon paste electrode (M-Chs NC/CPE). The reviewer feels that not only the researchers involved in sensors but also many readers interested in material chemistry may get inspiration from this report. The reviewer recommends the publication of this manuscript after minor revisions.
- In Figure 3b, the peak located at around 21oshould be clarified. It seems the peak also shows a positive shift. Please provide some explanination.
Thanks for the reviewer for his comment, this peak was assigned to the extracred chitosan from shrimb shell which was bascillay located at 20.01º in the XRD pattern of pure chitosan. However, the peak was shifted to 21o and became broad and less intense after capping of Fe3O4 with chitosan and hence it can be clearly stated that the particle size decreased after coating procedure. The coating of amorphous chitosan on crystalline Fe3O4 may induce microstrain which resulted in broadening of peaks in case of coated sample (https://doi.org/10.1016/j.apsusc.2013.09.169). This shift of the peak position and the less intensity were described by many pervious works such as (https://doi.org/10.1021/acssuschemeng.8b04028). This shift was attributed to the effect of introduction of cations into the chitosan matrix with the amino and hydroxyl groups destroying the orientation of the chitosan chains by the intermolecular and intramolecular H-bonding effect (https://doi.org/10.1016/S1005-9040(06)60214-6).
- In Figure 4, the font size of the scale bar is not unified.
The Reviewer's remark is completely right, but we think that the scale bar number put inside each picture may be better due to the differences between SEM and TEM results.
- In Figure 6a, the peak intensity and positions of cathodic peaks are also recommended to be described.
A heartfelt thanks to the reviewer for his valuable recommendation,it was described in the revised manuscript.
I will very much appreciate receiving any other comments again. Any other suggestions are mostly welcome and please accept all my best wishes.
Sincerely Yours
Sayed H. Kenawy
College of Science,
Chemistry Department,
Imam Mohammad Ibn Saud Islamic University (IMSIU),
Riyadh 11623, KSA
E-mail: skibrahim@imamu.edu.sa

Round 2
Reviewer 1 Report
accept